# Unveiling the Properties of Thai Stingless Bee Propolis via Diminishing Cell Wall-Associated Cryptococcal Melanin and Enhancing the Fungicidal Activity of Macrophages

**DOI:** 10.3390/antibiotics9070420

**Published:** 2020-07-17

**Authors:** Ketsaya Mamoon, Patcharin Thammasit, Anupon Iadnut, Kuntida Kitidee, Usanee Anukool, Yingmanee Tragoolpua, Khajornsak Tragoolpua

**Affiliations:** 1Division of Clinical Microbiology, Department of Medical Technology, Faculty of Associated Medical Sciences, Chiang Mai University, Chiang Mai 50000, Thailand; ketsaya_m@cmu.ac.th (K.M.); patcharin_th@cmu.ac.th (P.T.); anupon_i@cmu.ac.th (A.I.); usanee.anukool@cmu.ac.th (U.A.); 2The Graduate School, Chiang Mai University, Chiang Mai 50000, Thailand; 3Center for Research and Innovation, Faculty of Medical Technology, Mahidol University, Bangkok 10100, Thailand; kuntida.kit@mahidol.ac.th; 4Infectious Diseases Research Unit (IDRU), Faculty of Associated Medical Sciences, Chiang Mai University, Chiang Mai 50000, Thailand; 5Department of Biology, Faculty of Science, Chiang Mai University, Chiang Mai 50000, Thailand; yingmanee.t@cmu.ac.th

**Keywords:** stingless bee propolis, *Cryptococcus neoformans*, melanin, chitin-chitosan, phagocytosis

## Abstract

*Cryptococcus neoformans*, a life-threatening human yeast *pathogen,* has the ability to produce melanin, which is one of the common virulence factors contributing to cryptococcal pathogenesis. This virulence factor is closely associated with the cryptococcal cell wall, specifically chitin and chitosan polysaccharides, a complex structure that is essential for maintaining cellular structure and integrity. In this study, we aim to investigate the effects of two stingless bee (SLB) propolis from *Tetragonula laeviceps* and *Tetrigona melanoleuca* against cell wall-associated melanin in *C. neoformans,* and its immune response in RAW 264.7 macrophage. The ethanolic extract of SLB propolis (EEP) has strongly exhibited anti-cryptococcal activity. Moreover, EEP from both sources reduced chitin/chitosan and melanin production against *C. neoformans* in a dose-dependent manner. Likewise, the mRNA expression level of *CDA1, IPC1-PKC1* and *LAC1* genes involved in the cryptococcal melanization pathway was significantly decreased at 2 mg/mL in EEP treatment. Additionally, pretreatment with EEP prior to yeast infection dramatically reduced intracellular replication of *C. neoformans* in RAW 264.7 macrophages in a dose-dependent manner. This study might be a new insight to use a natural powerful source, not only acting to target cell wall-associated molecules, but also being capable to explore a novel strategy by which dysregulation of these molecules leads to promote immunomodulatory activity.

## 1. Introduction

Cryptococcosis is an opportunistic fungal infection caused by *Cryptococcus neoformans*, an encapsulated yeast that has been a major public global health problem worldwide. The main parts of the cell wall-associated melanin of *C. neoformans* are composed of glucans, chitin and melanin. The α-1, 3-glucans in *C. neoformans* are required to anchor the polysaccharide capsule to the cell wall [1]. In addition, Ricardo et al. has described the process of cell wall melanization, requiring additional components for the attachment to melanin, specifically chitin, chitosan and glucans in the cell wall [2]. Chitin, a β-(1, 4)-linked polymer of N-acetylglucosamine (GlcNAc), is one of the most abundant biopolymers and is essential to the integrity of the septum and fungal cell wall. Chitosan, the deacetylated derivative form of chitin, is produced enzymatically by the chitin deacetylases encode *CDA1* gene. Deacetylation of chitin is catalyzed by chitin deacetylases (Cdas) and produces chitosan, also an important component of the cell wall at various times during the life cycle. Chitin/chitosan absence also affects two major virulence factors, melanin and capsule, of *C. neoformans* [3].

Melanin is synthesized with the presence of 2,3- or 3,4-diphenol substrates such as dopamine, epinephrine, norepinephrine and 3,4-dihydroxyphenylalanine (DOPA), resulting in the production of a black pigment. The *LAC1* genes encode the laccase enzyme which catalyzes the formation of melanin by oxidizing the substrate and accumulating in the cell wall of *C. neoformans* [4]. Recently, the role of the Inositol-Phosphoryl Ceramide synthase 1- Diacylglycerol-Protein Kinase C1 (Ipc1-DAG-Pkc1) pathway has contributed to the cell wall melanogenesis and has provided evidence that signal transduction has occurred through this pathway [5]. Cell wall-associated melanin, located in a position to interact with free radical-generating substances, such as reactive oxygen species (ROS) and reactive nitrogen species (RNS) of phagocytic cells, play a key role in protecting *Cryptococcus* yeast cells from the host [6]. Therefore, the localization of melanin is a key step in the host’s defense to modulate the interaction of *C. neoformans* with phagocyte cells. Melanization serves to prevent the microorganism from oxidative stresses within the phagolysosome by breaking down host substrates, resulting in intracellular replication. Their ability assists yeast cells to escape immune cells and macrophages, and lead the way without being killed [7].

Nowadays, a regimen for cryptococcosis treatment by the polyene macrolides, amphotericin B, has been widely used. However, amphotericin B is able to induce acute nephrotoxicity by comprehensive gene expression changes in the mouse model [8]. Unfortunately, a killing assay demonstrated that *C. neoformans* melanin contributes to a decrease in susceptibility to amphotericin B [9], owing to the fact that limitations are less effectively resolved by antifungal properties and high-toxicity therapies. Thus, it is extremely important to find the agent of control for these harmful effects. Lately, there has been extensive research focusing on the exploration of natural substances to target fungal virulence factors that encourage macrophage polarization to M1-macrophages and eliminate the pathogen by themselves.

Propolis, or bee glue, is a complex mixture of materials collected from a variety of bee species. Bees use propolis to seal cracks, maintain stable temperature and humidity levels in a bee hive, as well as protect larvae from microorganisms, e.g., bacteria and fungi [10]. The major chemical compositions in propolis are flavonoids, phenolic acids and terpenes [11]. Sanpa et al. demonstrated that the stingless bee (SLB) propolis (*Tetragonula laeviceps*) inhibited the growth of *Staphylococcus epidermidis* [12]. Recently, Thammasit et al. reported the potential of propolis from *Apis mellifera* to decrease the major virulence factors of *C. neoformans* [13]. However, the effect of SLB propolis interfering with cell wall-associated melanin retention in the cell wall of *C. neoformans*, and the immune responses of phagocytes in the defense against SLB propolis-treated yeast infections, has not been previously reported.

This study is the first report of the translational potential of SLB propolis from *Tetragonula laeviceps* (Chantaburi) and *Tetrigona melanoleuca* (Chiang Mai) to alter the state of chitin/chitosan and stimulate melanization of *C. neoformans*, resulting in the promotion of phagocytosis and the decrease of intracellular replication of yeasts in macrophages. Therefore, SLB propolis exhibits promise for an alternative treatment, focused not only on the anti-virulence factors against *C. neoformans*, but also on the ability to synergistically modulate the immune response of the host.

## 2. Materials and Methods 

### 2.1. Yeast Cells and Culture Conditions

*C. neoformans* H99 was kindly provided by Assoc. Prof. Pojana Sriburee (Department of Microbiology, Faculty of Medicine, Chiang Mai University, Chiang Mai, Thailand). Yeast cells were maintained on Sabouraud Dextrose Agar (SDA) (HiMedia, India) and incubated at 30 °C, for 48–72 h. The isolated colony was picked and cultured in Sabouraud Dextrose Broth (SDB) (HiMedia, India) at 37 °C and shaken for 18–24 h.

### 2.2. Propolis Preparation

#### 2.2.1. Extraction of Ethanolic Extract of SLB Propolis (EEP)

*Tetragonula laeviceps* propolis from Chantaburi province (the eastern region of Thailand) and *Tetrigona melanoleuca* propolis from Chiang Mai province (the northern region of Thailand) were extracted via the maceration method, as described elsewhere [14]. One hundred grams of raw propolis was frozen at −80 °C overnight, ground, and homogenized with 70% ethanol prior to extraction. The solution was macerated for 72 h in a dark environment. After maceration, the solution was filtered by Whatman filter paper no. 1 and kept at 4–8 °C overnight for wax removal. The filtrate was concentrated by means of evaporation for the removal of the organic solvent in a rotary evaporator (Schott-DURAN, Germany). The ethanolic extract of SLB propolis (EEP) was lyophilized, dissolved in dimethyl sulfoxide (DMSO) and stored at −20 °C until use. DMSO was used as a vehicle control for all of experiments.

#### 2.2.2. Total Phenolic Content

Dissolved EEP was mixed with absolute methanol at an optimal concentration. Then, the solution was mixed with distilled water, ethanol and 50% Folin-Ciocalteu’s reagent. After incubation at room temperature for 5 min, 5% sodium carbonate was added, and the mixture was incubated at room temperature for 1 h in a dark environment. The reaction absorbance was measured at a wavelength of 725 nm using an Enzyme-Linked Immunosorbent Assay (ELISA) microplate reader (BioTek, VT). The total phenolic compound was calculated from the gallic acid standard curve [14].

#### 2.2.3. Total Flavonoid Content

Dissolved EEP was mixed with absolute methanol at an optimal concentration. Then, methanol, 10% aluminum chloride, potassium acetate and distilled water were added, and the mixture was incubated at room temperature for 30 min. The absorbance was measured at a wavelength of 415 nm using the ELISA microplate reader. The total flavonoid content was calculated with the quercetin standard curve [14].

#### 2.2.4. High-Performance Liquid Chromatography (HPLC) Analysis

Analytical HPLC was run on the HPLC system (Agilent technologies 1200 series, Santa Clara, CA, USA) (separation was achieved on a 2.1 × 50 mm, 1.8 µm threaded column, Agilent Eclipse XDB-C18, 4.6 × 150 mm, 5 µm particle size). The mobile phases consisted of 0.1% formic acid in deionized water (solvent A) and methanol (solvent B). The elution was carried out with a linear gradient and a flow rate of 1.0 mL/min. The detection was monitored at 267 nm and the components were identified in comparison with commercial standards (Gallic acid, Quercetin, Pinocembrin, Chrysin, and Galangin) [15]. All standards were purchased from Sigma-Aldrich, St. Louis, MO, USA.

### 2.3. Effect of EEP on Chitin/Chitosan Production of C. neoformans

#### 2.3.1. Chitin and Chitosan Synthesis

*C. neoformans* was treated with EEP at a concentration of 0.5, 1 and 2 mg/mL for 2 h at 37 °C. EEP-treated *C. neoformans* was washed with 1× PBS and incubated with calcofluor white (CFW) (Sigma-Aldrich, St. Louis, MO, USA), which binds to chitin in the fungal cell walls, at 1 µg/mL for 30 min. After washing, the pellets were suspended with 1× PBS and the chitin was detected. For chitosan detection, 50 µg/mL of eosin Y (EY) (Merck, Darmstadt, Germany) were added to EEP-treated *C. neoformans* and incubated for 15 min at room temperature. The cells were washed with McIlvaine’s buffer (0.2 M of Na_2_HPO_4_ and 0.1 M of Citric acid, pH 6.0) and the pellets were suspended with washing buffer [16]. The fluorescence intensity was measured by a fluorescence microplate reader (BioTek, Winooski, VT, USA) (wavelengths were set at 355/433 nm for CFW and 488/548 nm for EY).

#### 2.3.2. Detection of CDA1 mRNA by Real-Time Reverse Transcription-Polymerase Chain Reaction (rRT-PCR)

The expression of *CDA1* mRNA, which encodes the chitin deacetylase enzyme that converts chitin to chitosan, was determined by rRT-PCR. The total RNA was extracted from lysed yeast cells using TRIZOL^®^ reagent (Invitrogen, CA) according to the manufacturer’s instructions. The RNA yield and the purity were measured using a nanodrop spectrophotometer (BioTek, Winooski, VT, USA) at optical density (OD) 260/280 nm. Five hundred nanograms of total RNA were reverse transcribed to cDNA using RevertAid First Strand cDNA Synthesis Kit (Thermo Fisher Scientific, Waltham, MA, USA). The *CDA1* and *ACT1* were performed by SYBR Green qPCR Master Mix (Thermo Fisher Scientific, Waltham, MA, USA) and specific primers. The PCR primer sequences were designed according to the *CDA1* and actin (*ACT1*). *CDA1* (accession no CNAG_05799, 180 bp; forward: 5′-CTTCTTACACTGATGGCTCAAC-3′; reverse: 5′-CAACACTCTGCTGGTAGATGTC-3′); and *ACT1* (accession no CNAG_00483, 136 bp; forward: 5′-CCTTGCTCCTTCTTCTAT-3′; reverse: 5′-CTCGTCGTATTCGCTCTT-3′). The conditions were performed in 35 cycles: initial denaturation at 94 °C for 30 s, annealing at 58 °C for 30 s and extension at 70 °C for 60 s, followed by cooling at 37 °C for 30 s. The expression of the target mRNA level was analyzed by the 2^−∆∆CT^ method and expressed as relative fold change when normalized with *ACT1* as a housekeeping gene [17].

### 2.4. Effect of EEP on C. neoformans Melanization

#### 2.4.1. Melanin Production 

*C. neoformans* was treated with EEP at concentrations of 0.5, 1 and 2 mg/mL for 2 h at 37 °C [18]. Yeasts were then cultured in the L-3,4-dihydroxyphenylalanine (L-DOPA) minimal medium (15 mM glucose, 10 mM MgSO_4_, 29.4 mM KH_2_PO_4_, 13 mM glycine and 3 mM thiamine; pH 5.5), with 1 mM L-DOPA (Sigma-Aldrich, St. Louis, MO, USA) and incubated in a dark chamber at 30 °C and in a shaking incubator for 5 days. The melanin pigment production was observed daily. To confirm the viability of yeast cells, the cells were diluted, adjusted to 5 × 10^5^, 5 × 10^4^ and 5 × 10^3^ cells/mL, and dropped on a SDA plate. After 48 h of incubation at 37 °C, the number of colony-forming units (CFUs) were counted [19]. 

#### 2.4.2. Laccase Activity

The EEP-treated *C. neoformans* was adjusted to 1.2 × 10^7^ cells and grown in 0.01% L-DOPA minimal medium and incubated at 30 °C on 250 rpm shaking. After 16 h of incubation, the cultures were incubated at 25 °C for 24 h while shaking. The 1 mL of supernatant was collected at various times and was then centrifuged. The enzymatic activity was defined as 1 U equal to 0.0001 AU spectrophotometrically read at 475 nm [18].

#### 2.4.3. Detection of Melanin-Related Gene Expression by rRT-PCR

The *LAC1* (encoding the laccase enzyme) was detected by rRT-PCR, and further confirms the transcriptional melanin-related gene, *IPC1* (encoding inositol-phosphoryl ceramide synthase 1) and *PKC1* (encoding protein kinase C1). The steps for cDNA preparation and rRT-PCR were performed in the previous section. The specific primers (Appendix A) were used for *LAC1*, *IPC1* and *PKC1* mRNA. The expression of the target mRNA level was analyzed by the 2^−∆∆CT^ method using *ACT1* as a housekeeping gene.

### 2.5. Macrophages Cultures 

The RAW 264.7 murine macrophage cell line was cultured in Dulbecco’s minimal essential medium (DMEM) (GIBCO, CA) and supplemented with 10% heat-inactivated fetal bovine serum (FBS) (GIBCO, CA), 4 mM L-glutamine, 100 Units/mL of penicillin and 100 µg/mL of streptomycin. The cells were maintained in a humidified environment of 5% CO_2_ at 37 °C.

### 2.6. Immune Response with EEP-Treated C. neoformans

#### 2.6.1. Macrophages Infected with EEP-Treated C. neoformans

The RAW 264.7 macrophages were maintained in a cell culture plate and induced to classical activated macrophages by adding 0.6 µg/mL of lipopolysaccharide (LPS) and 100 ng/mL of interferon-γ (IFN- γ), and incubated at 37 °C in a 5% CO_2_ humidified incubator for 24 h [20]. EEP-treated *C. neoformans* was opsonized with 1:10 of anti-Glucuronoxylomannan (GXM) monoclonal antibody (Clone 18b7) for 1.30 h 37 °C in a 5% CO_2_ atmosphere. Macrophages cells were infected with 5 multiplicity of infection (MOI) of opsonized EEP-treated *C. neoformans* and incubated at 37 °C in 5% CO_2_ for 2 h. For the experiment of intracellular cryptococcal cell replication, the EEP-treated *C. neoformans* was stained with CFW before opsonization. 

#### 2.6.2. Intracellular Cryptococcal Cell Replication 

The activated-RAW 264.7 cells were infected with EEP-treated *C. neoformans* and labeled with CFW, as described above. After incubation, the supernatant was removed and continuously incubated at 37 °C in a 5% CO_2_ atmosphere for 24 h in fresh medium. The cells were lysed using lysis medium (0.01% bovine serum albumin (BSA) and 0.01% Tween-80 in distilled water) and the intracellular yeasts were counted and expressed in colony-forming units (CFUs) [21].

#### 2.6.3. LysoTracker Staining 

The RAW 264.7 cells were infected with EEP-treated *C. neoformans* and labeled with CFW, as described above. After incubation, the supernatant was removed and stained with LysoTracker Red (Thermo Fisher, Waltham, MA, USA) at 37 °C in a 5% CO_2_ atmosphere for 1 h in fresh medium according to the manufacturer’s instructions. The cells were washed twice in sterile PBS. Then, the infected cells were observed for yeast and phagolysosome localization under an inverted fluorescence microscope (Zeiss, Oberkochen, Germany).

### 2.7. Statistical Analysis

All data were represented as the mean ± standard error of the mean (SEM) of three independent experiments. Statistical analyses were carried out using a one-way analysis of variance (ANOVA). Values of *p* < 0.05 were considered significant.

## 3. Results and Discussion

### 3.1. Quantitation of Essential Compounds in SLB Propolis

The total phenolic compounds of EEP from Chiang Mai and Chantaburi sources were calculated as 4.31 ± 0.11 and 3.84 ± 0.03 mg gallic acid equivalents (GAE)*/*g extract, respectively. The flavonoid compound from EEP was founded on 6.53 ± 1.07 and 4.75 ± 1.56 mg quercetin equivalents (QE)*/*g extract, as shown in Table 1. The chemical compounds in EEP were analyzed by HPLC. Chromatogram profiles exhibited peak readings and were calculated in comparison to standard agents (Figure 1a–c). This analysis also demonstrated the quantification of the indicated bioactive compounds in EEP (Table 2). The EEP from Chiang Mai was comprised of gallic acid, quercetin and pinocembrin as 0.34, 1.13 and 2.19 µg/mL, respectively. In comparison with the standard agents, only gallic acid was found in the EEP from Chantaburi at a concentration of 1.03 µg/mL. Galangin was not found in EEP from both sources. The chemical composition varies due to different regions, plant species used to produce propolis by bees, time of collection [22] and extraction method [23]. The genus Melipona and Trigona are the two major genera of stingless bees in India. Indian stingless bee propolis has characteristically been found with gallic acid and naringin, a type of unique propolis, which may also contain other compounds such as sugars, terpenes and steroids [24]. Conversely, Australian stingless bee propolis, belonging to the genera Tetragonula and Austroplebeia, is found to be rich in diterpenic acids (pimaric and isopimaric acids) and gallic acids [25]. Therefore, it is certain that propolis produced by different species of stingless bees and the resin from plant origins will contain different chemical compounds.

### 3.2. Effect of EEP on Chitin/Chitosan Synthesis of C. neoformans

In this study, we evaluated the effects of SLB propolis on cell viability by trypan blue assay. All concentrations of EEP did not have an effect on the growth of *C. neoformans*. However, EEP at a concentration of 4 and 8 mg/mL exhibited clumping. Thus, these evidences and the high concentration of DMSO need to be taken into account (Appendix A). The chitin content of EEP-treated *C. neoformans* was assessed by mean fluorescence intensity (MFI) of CFW. As shown in Figure 2a, the two sources of EEP-treated *C. neoformans* led to a significant decrease in chitin content (*p* < 0.05) in a dose-dependent manner, contrasting with the vehicle-treated yeasts where DMSO did not alter the chitin. Furthermore, we observed a remarkable decrease of *CDA1* mRNA expression on EEP-treated *C. neoformans* at a concentration of 1 and 2 mg/mL of both EEPs (Figure 2b). The chitosan level was detected by EosinY staining. The results were in agreement with the chitin levels of EEP-treated *C. neoformans* in a dose-dependent manner. Particularly, EEP-Chiang Mai at a concentration of 2 mg/mL have significantly reduced the chitosan content of *C. neoformans* by approximately 60% when compared with the vehicle control, as shown in Figure 2c. These results indicate that EEP can interfere with the enzymatic pathway and include some genes associated with enzymatic modification of chitin/chitosan biosynthesis. The decrease in chitin might be caused by phenolic acid, a major constituent of propolis, by disrupting the chitin synthesis process. Naturally, chitin synthases (CHS), which polymerizes GlcNAc into chitin from cytoplasmic pools of uridine diphosphate N-acetylglucosamine (UDP-GlcNAc), are integrated into the plasma membrane by multiple trans-membrane helices [26]. The chelating property of flavonoids present in stingless bee propolis has been investigated with transitional metal ions chelators, such as iron (Fe^2+^), cobalt (Co^2+^), nickel (Ni^2+^), copper (Cu^2+^) and zinc (Zn^2+^) [27]. In a previous study, the quercetin-metal complexes demonstrated an interaction with many kinds of enzymatic activities [28]. It has been well established that the chitosan biosynthesis requires the chitin deacetylase (CDA) to convert chitin into chitosan as a structural component of the cell wall. Chitin deacetylase has been recommended to be a metalloenzyme and its catalytic ability can be highly influenced by divalent cations. Moreover, specifically the activity of CDA could be enhanced in the presence of calcium and cobalt ions [29]. Strains of *C. neoformans* lacking *CDA1* have significantly reduced chitosan and are sensitive to cell wall inhibitors [30]. It is particularly interesting to note that the reduction of chitosan on the cell wall of EEP-treated *C. neoformans* is being considered to be due to the enzyme that was chelated by essential composition in the stingless propolis.

### 3.3. Anti-Melanization of EEP-Treated C. neoformans

The effects of EEP on melanin production were examined in phenotype appearance in broth medium, and the results confirm the reduction of melanin production with laccase activities, also seen in mRNA expression. The results indicate that both EEPs decrease the melanin production in *C. neoformans* in a dose-dependent manner, compared to the growth control and vehicle control, as illustrated by the color transition of a shade of black to light brown color (Figure 3a). The viable EEP-treated yeast cells in L-DOPA medium confirmed the growth by drop plate technique after 5 days of incubation. This result indicated an equal amount of EEP-treated yeast cells compared to the growth control and vehicle control, as shown in Figure 3b. Thus, the reduction of melanin synthesis was caused by the effect of EEP and was not involved in the amount of yeast. EEP-Chiang Mai, at a concentration of 2 mg/mL, has significantly reduced the laccase activity of *C. neoformans*. Surprisingly, all concentrations of EEP-Chantaburi have also significantly decreased the laccase activity (*p* < 0.05) of EEP-treated *C. neoformans* in comparison with the growth and vehicle controls (Figure 3c). In addition, the laccase-related gene, *LAC1*, was also determined. The result found that *LAC1* mRNA expression was attenuated and presented a significant reduction at 0.5-fold changes in 2 mg/mL of EEP from Chiang Mai-treated *C. neoformans*, compared to the vehicle control (Figure 4a). Interestingly, EEP from Chantaburi-treated *C. neoformans* has shown a greater reduction of *LAC1* mRNA expression, in a range of 0.5- to 0.8-fold changes in a dose-dependent manner. The transcription factor of a melanin-related synthesis pathway (PKC regulation pathway) was investigated. The results showed that both sources of EEP at a concentration of 2 mg/mL dramatically decreased the expression of *IPC1* and *PKC1*, as illustrated in Figure 4b,c. These results indicate that EEP contributes to the regulation of melanogenesis by targeting the Ipc1-Pkc1-laccase cascade as a regulator of the virulence factor of *C. neoformans*. Cryptococcal melanin is synthesized by laccase, a secretion of copper-dependent oxidases which requires metal ions in an active site, the copper ion binding regions of the laccase protein [31]. Silva et al. investigated the copper-chelating properties of microplusin. Microplusin was able to inhibit melanization and reduce the laccase activity of *C. neoformans* strain H99 [32]. Most considerably, the strain IPC1/ΔC1-PKC1 produced less laccase activity and melanin pigment compared to the parental strain H99 [5]. Accordingly, the SLB propolis restricts the function of IPC1 and PKC1 by interfering with the phosphotransferase activity of the kinase enzyme. Likewise, quercetin and 16 related flavonoids act as inositol phosphate kinase inhibitors by competing for their adenosine triphosphate (ATP)-binding sites [33].

### 3.4. Intracellular Killing of EEP-Treated C. neoformans in Phagolysosome in Macrophages

The responses of the macrophages with EEP-treated *C. neoformans* were investigated with phagocytosis activity assay and intracellular cryptococcal proliferation. The RAW 264.7 macrophage cells were infected with EEP-treated *C. neoformans*, stained with Wright-Giemsa (Figure 5a), and the phagocytosed cells were counted. The results showed that the percentage of phagocytosis was between 6% and 10% in both sources of EEP-treated *C. neoformans* in every time period (Appendix A), while the phagocytosis index was 2–3 cells/macrophage in both EEP sources (Appendix A). The results indicated that there was no difference in phagocytosis activity in EEP-treated *C. neoformans* and the control group.

The intracellular cryptococcal proliferation was stained with CFW and photographed with a fluorescence microscope. Mother yeast cells displayed a bright fluorescent intensity, while budding-daughter cells had a weaker fluorescent intensity compared to the mother cells, as shown in Figure 5b. The results from the confocal laser scanning microscope (CLSM) showed clear and reproducible defects in intracellular survival of EEP-treated *C. neoformans* in RAW 264.7 macrophage when compared with vehicle control (Figure 5c).

We hypothesized that its susceptibility to oxidative and nitrosative stress would affect the survival of EEP-treated yeast cells within phagocytes. Thus, the presence of intracellular yeast was confirmed by phagolysosomal trafficking assay. EEP-treated yeast cells were stained with CFW, a blue fluorescent dye that labels chitin in the cell wall and opsonized with anti-GXM mAb (Clone 18b7) prior to infection. The intracellular yeast cells within phagolysosome were observed by LysoTracker Red, an acidophilic fluorescent dye that labels lysosomes. The result signified the labeled yeasts located in phagolysosome of RAW 264.7 macrophage cells as shown in Figure 5d. Further, we determined the survival of internalized yeast cells inside 264.7 macrophage cells by using CFU assay. Interestingly, both sources of EEP at a concentration of 2 mg/mL have been shown significantly decrease the growth of *C. neoformans* by approximately 20% of CFU (0.8-fold changes) compared with the vehicle control group (Figure 5e). There were no alterations in either EEP-treated group or the control group in terms of cryptococcal uptake by macrophages. In this context, it is interesting to note that SLB propolis did not affect the capsular production of cryptococci (data not shown). Thus, both groups (EEP-treated group and control group) of opsonized-cryptococci presented a similar percentage of phagocytosis and phagocytosis index. Taken together, we speculated that the reduced intracellular survival of EEP-treated *C. neoformans* at a high concentration was due to the decreased melanin pigment. Consequently, they were susceptible to oxidative damage, presenting a significant defect in survival in RAW 264.7 macrophages, as shown in Figure 5e. Indeed, the concentration of EEP that was used in this experiment was low and was insufficient to compete reactive oxygen and nitrogen species (ROS/RNS) as scavengers. Thus, the results of survival from assay were from ROS/RNS of macrophages. As previously mentioned, melanized *C. neoformans* strains were less susceptible to nitrosative and oxidative stresses than melanin-deficient strains [34]. Moreover, Steenbergen et al. compared the survival of melanized and non-melanized cells from the *C. neoformans* strain Cap67 (acapsular strain) and found that the melanized cells were significantly more resistant to being killed by Acanthamoeba castellanii [35]. In the environment, *C. neoformans* have a small capsule [36] and can be melanized [37]. Thus, we suggest that defects of the cell wall-melanin in *C. neoformans* by EEP decreases cryptococcal intracellular survival and it is deemed susceptible to being killed by the reactive molecules of macrophage cells.

## 4. Conclusions

In summary, this study is the first report on the biological activities of *Tetrigona melanoleuca* propolis and *Tetragonula laeviceps* propolis against *C. neoformans*. The data presented here show that the chemical composition of SLB propolis is complex and depends on the species. Both of the SLB propolis have effectively reduced chitin/chitosan-associated melanin pigment through a Ipc1-DAG-Pkc1-Lac1 pathway, which involved the disturbance of cryptococcal melanization. Importantly, interfering with this pathway impairs the ability of yeast cells to proliferate within macrophages, causing a response to oxidative stress and increased degradation in phagolysosome. The current findings suggest that SLB propolis may be a potential candidate for an anti-virulence agent and could be developed for cryptococcal treatment.

## Figures and Tables

**Figure 1 antibiotics-09-00420-f001:**
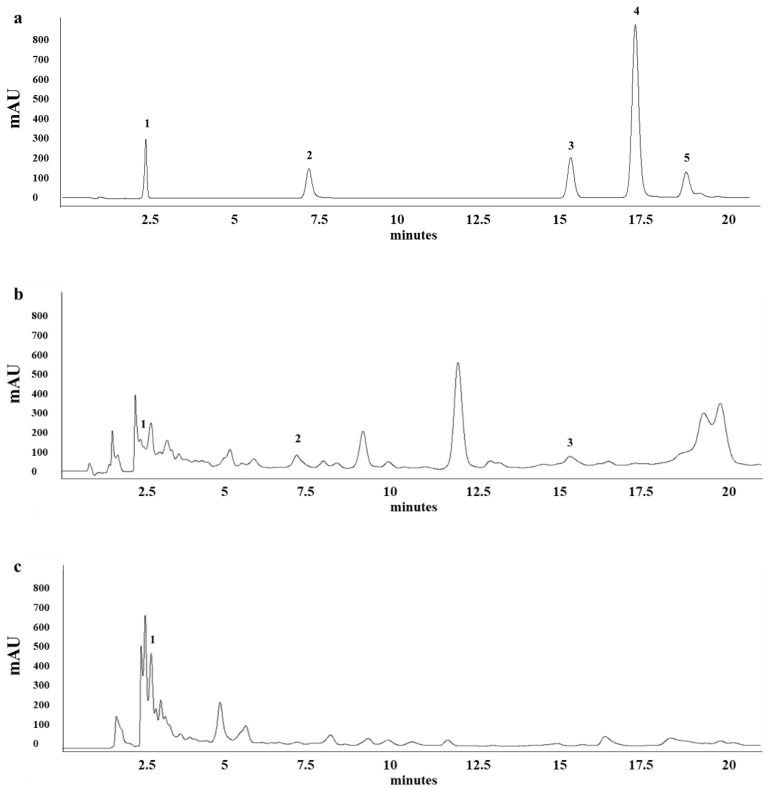
High-performance liquid chromatography (HPLC) chromatograms of EEP consistent with the quantitation and identification. (**a**) The standard composition: (1) gallic acid, (2) quercetin, (3) pinocembrin, (4) chrysin, and (5) galangin. (**b**) EEP of Chiang Mai. (**c**) EEP of Chantaburi.

**Figure 2 antibiotics-09-00420-f002:**
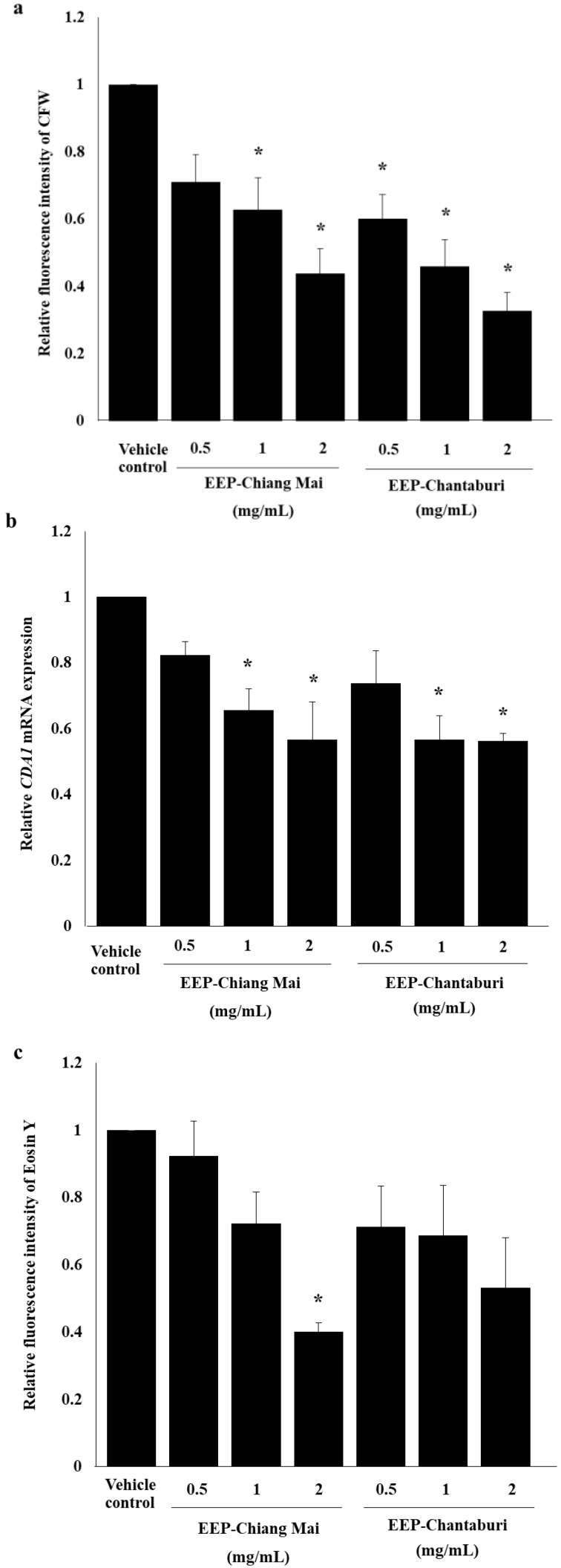
Effect of EEP on chitin/chitosan synthesis of *C. neoformans*. EEP-treated *C. neoformans* were incubated for 2 h. (**a**) The chitin content was determined by CFW staining and calculated to relative fluorescence intensity. (**b**) *CDA1* mRNA expression by rRT-PCR. (**c**) The chitosan production was measured by EosinY staining. Significant differences (* *p* < 0.05) were determined by one-way ANOVA and compared with the vehicle control.

**Figure 3 antibiotics-09-00420-f003:**
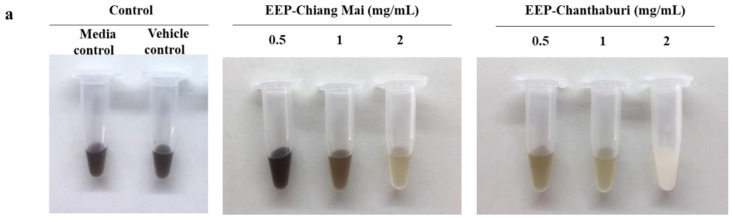
Anti-melanization of EEP-treated *C. neoformans*. (**a**) Melanization of *C. neoformans* in the presence of L-DOPA at various concentrations of EEP for 5 days of incubation. (**b**) Colony-forming units (CFU) of EEP-treated *C. neoformans* on SDA. (**c**) Laccase activity was investigated after the EEP treatment. The results are expressed as mean ± SEM of three independent experiments in duplicate. Significant differences (* *p* < 0.05) when compared with the vehicle control were determined by one-way ANOVA.

**Figure 4 antibiotics-09-00420-f004:**
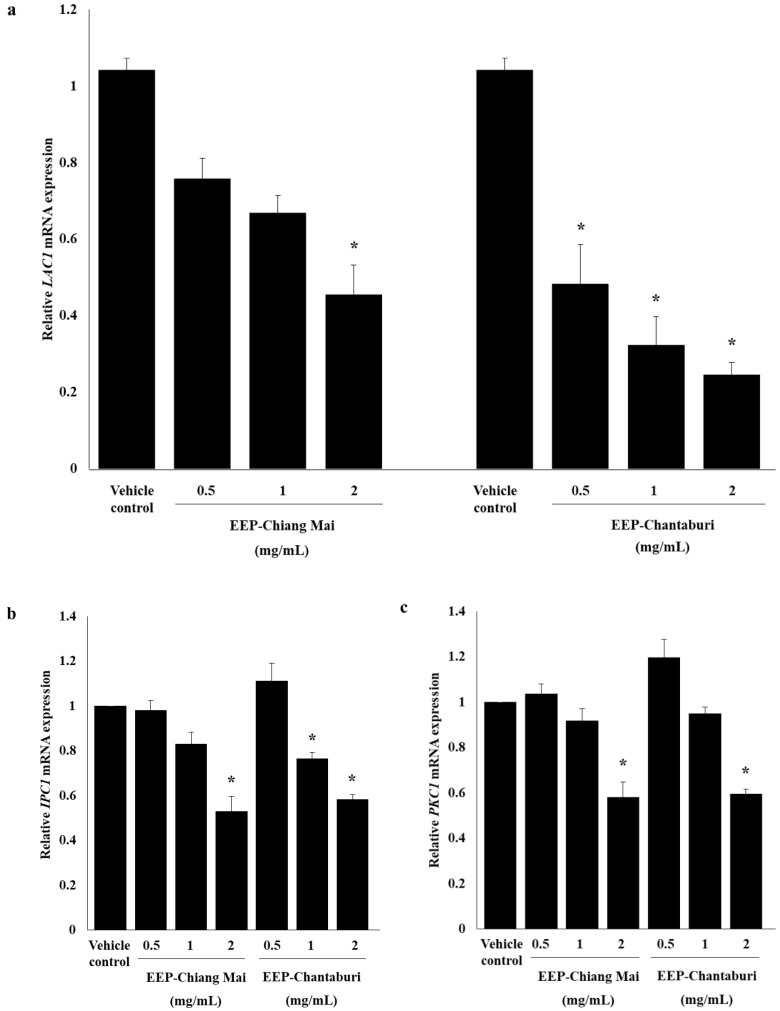
Melanogenesis-related genes mRNA expression of EEP-treated *C. neoformans*. (**a**) *LAC1* gene-related melanin production was determined and normalized with the housekeeping gene level and repressed as a fold change compared to vehicle control. (**b**,**c**) IPC1 and PKC1 pathway-related melanin production was detected and repressed as a fold change compared to vehicle control. The results are expressed as mean ± SEM of three independent experiments in duplicate. Significant differences (* *p* < 0.05) when compared with the vehicle control were determined by one-way ANOVA.

**Figure 5 antibiotics-09-00420-f005:**
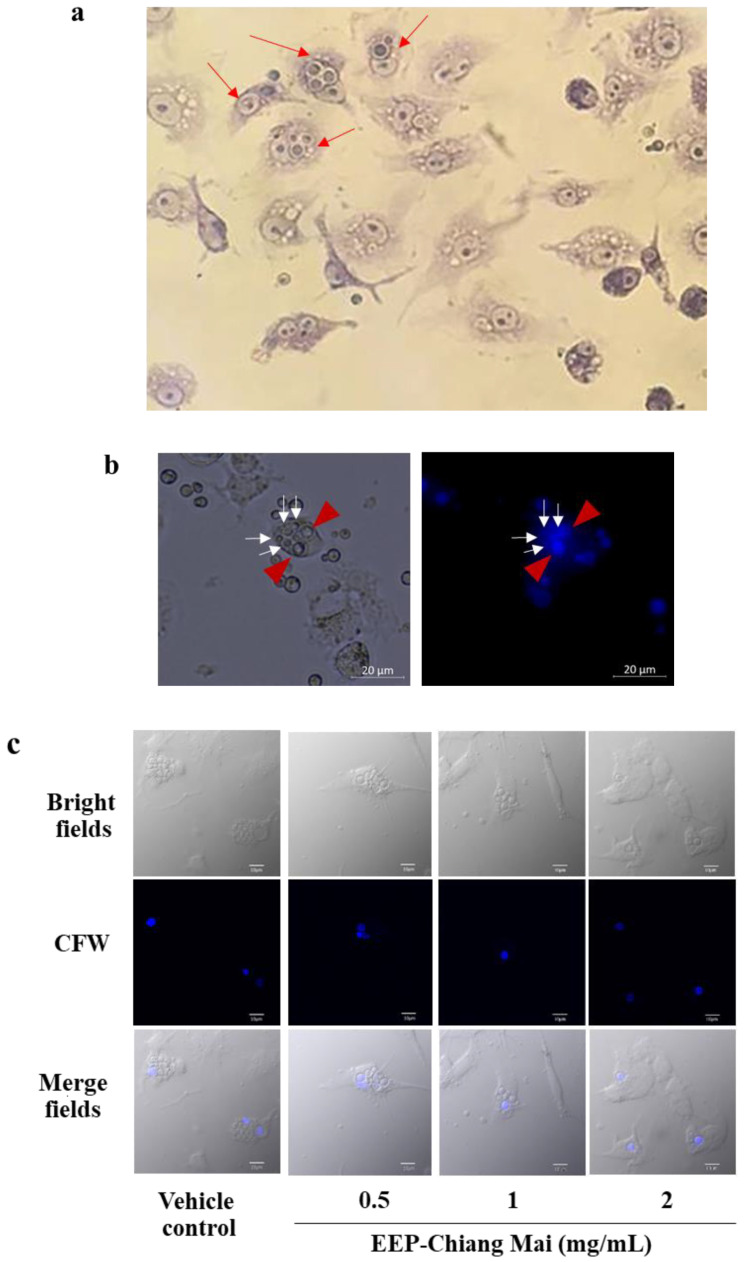
Anti-proliferative effect of EEP on *C. neoformans* phagocytosis by RAW 264.7 macrophages. EEP-treated yeast cells were opsonized with anti-GXM mAb (Clone 18b7) prior to infection for various time points. (**a**) Infected macrophages with *C. neoformans* were stained with Wright-Giemsa staining. (**b**,**c**) Intracellular proliferation of untreated and EEP-treated *C. neoformans* after infection for 24 h. Mother cells—bright blue color (red arrow), daughter cells—dim color (white arrow). Scale bar represents 20 µm (magnification × 400). (**d**) Localization of *C. neoformans* to phagolysosomes. Infected macrophage cells were stained with LysoTracker Red and observed under inverted fluorescence microscope. Scale bars represent 20 µm. (**e**) Intracellular survival of EEP-treated *C. neoformans*. Macrophage cells were lysed using lysis medium and intracellular cryptococcal proliferation were counted by colony-forming units (CFUs) and shown as fold change. The data represent the mean ± standard error of the mean (SEM) of at least three independent experiments. Significant differences (* *p* < 0.05) when compared with the vehicle control were determined by one-way ANOVA with the Tukey-Kramer multiple comparisons post-test.

**Table 1 antibiotics-09-00420-t001:** Total phenolic and flavonoid contents of ethanolic extract of SLB propolis (EEP).

EEP Source	Total Phenolic (mg GAE ^1^/g extract)	Total Flavonoid (mg QE ^2^/g extract)
Chiang Mai	4.31 ± 0.11	6.53 ± 1.07
Chantaburi	3.84 ± 0.03	4.75 ± 1.56

^1^ GAE: gallic acid equivalents; ^2^ QE: Quercetin equivalents.

**Table 2 antibiotics-09-00420-t002:** Results of high-performance liquid chromatography EEP.

EEP Source	Chemical Compound	Concentration (µg/mL)
Chiang Mai	Gallic acid	0.34
	Quercetin	1.13
	Pinocembrin	2.19
Chantaburi	Gallic acid	1.03

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
