# Peer review of "Unveiling the Properties of Thai Stingless Bee Propolis via Diminishing Cell Wall-Associated Cryptococcal Melanin and Enhancing the Fungicidal Activity of Macrophages"

_antibiotics, 2020, doi:10.3390/antibiotics9070420_

Round 1
Reviewer 1 Report
An interesting and well designed paper. The conclusions appears to support the data and in general the data is well presented.
I'd say that determining gallic acid from the samples would be difficult as there appears to be more than one peak at that retention time. Though this fact maybe useful in separating the different EEP's in the future.
I think the error bars on figs 4b-d are too large to make any firm conclusions on the number of phagocysed macrophages. I would remove them and state that the percentage of macrophages was between 6 and 10%.
I'm not absolutely convinced by the differentation of mother and daughter cells judging by the image but I'm sure the authors have looked at lots of experience looking at the images.
A few minor points
line 60 Cryptococcus should be italicised
line 103 how was the solution filtered?
Please give details about McIlvaine's buffer
it should be mL not ml
line 141 please bracket RT-PCR
line 173 just use RT-PCR rather than the full name
line 223 please put a space after the 4.31 and before 0.11 and for 3.84 and 0.03
No need to quote for 4 significant numbers in line 229
Genus names should be italicised
Author Response
Response to Reviewer 1 Comments
We sincerely appreciate the valuable comments and suggestions from the reviewers. We also thanks for giving us the opportunity to submit a revised of the manuscript. We will detail in our response below for a point-by-point response to the reviewers comments and concerns.
Point 1: I'd say that determining gallic acid from the samples would be difficult as there appears to be more than one peak at that retention time. Though this fact may be useful in separating the different EEP's in the future.
Response 1: We agreed with your suggestions about the chromatographic separation of gallic acid in EEP samples at a retention time of 2.5 minutes. The interference or other phytochemical compounds of EEP might be affect the retention and peak shape of gallic acid in EEP. Therefore, the optimized condition of wavelength, flow rate, mobile phase and stationary phase of gradient HPLC running should be concern in the future [1].
Point 2: I think the error bars on figs 4b-e are too large to make any firm conclusions on the number of phagocysed macrophages. I would remove them and state that the percentage of macrophages was between 6 and 10%.
Response 2: According to the reviewer’s suggestion, we agreed to remove figure 4b-e to the supplement data section and rewrote the clearly sentence in the revised manuscript at page 14 and line 480-481.
Point 3: I'm not absolutely convinced by the differentiation of mother and daughter cells judging by the image but I'm sure the authors have looked at lots of experience looking at the images.
Response 3: In fact, the macrophages were infected with calcofluor white-labeled C. neoformans. Before yeast replication, the intracellular yeast cells were brightly calcofluor white-fluorescent as assumed to be parent cells. While daughter cells budded from mother cells exhibited diluted the fluorescent dye leading to reduced fluorescence intensity (dim color) [2]. However, we also confirmed the proliferation of intracellular survival of C. neoformans in macrophages by colony forming units (CFUs) for represent in quantitative result.
Point 4: line 60 Cryptococcus should be italicized, line 103 how was the solution filtered?, Please give details about McIlvaine's buffer, it should be mL not ml, line 141 please bracket RT-PCR, line 173 just use RT-PCR rather than the full name, line 223 please put a space after the 4.31 and before 0.11 and for 3.84 and 0.03, No need to quote for 4 significant numbers in line 229, Genus names should be italicized.
Response 4: According to the reviewer’s suggestion, we corrected all of Point 4 in the revised manuscript.
- All of genus and species have been revised to italicized and including line 60 as well.
- We added the detail of the solution was filtered by Whatman filter paper no. 1 in the revised manuscript at page 3 and line 103-104.
- The contents of McIlvaine's buffer was added in the revised manuscript at page 4 and line 137-138.
- All of ml have been revised as mL.
- We rewrote the full name as real-time reverse transcription-polymerase chain reaction (rRT-PCR) in the revised manuscript at page 4 and line 141.
- We put the space after and before the numbers as your suggestion in the revised manuscript page 6 and line 214-217.
- The decimal number of the chemical compound concentration was rounded to 2 decimal places in the revised manuscript page 6 and line 220-221 and Table 2.
References
- Torres, A.R.; Sandjo, L.P.; Friedemann, M.T.; Tomazzoli, M.M.; Maraschin, M.; Mello, C.F.; Santos, A.S. Chemical characterization, antioxidant and antimicrobial activity of propolis obtained from Melipona quadrifasciata quadrifasciata and Tetragonisca angustula stingless bees. Braz J Med Biol Res, 2018, 51, e7118.
- Davis, M.J.; Eastman, A.J.; Qiu, Y.; Gregorka, B.; Kozel, T.R.; Osterholzer, J.J.; Curtis, J.L.; Swanson, J. A.; Olszewski, M.A. Cryptococcus neoformans-induced macrophage lysosome damage crucially contributes to fungal virulence. J Immunol, 2015, 194, 2219-31.

Reviewer 2 Report
The manuscript,antibiotics-847642, describes the effect of two Stingless Bee (SLB) propolis of Tetragonula laeviceps and Tetrigona melanoleuca on cell wall-associated melanin of C. neoformans and its immune response in RAW 264.7 macrophages.
This article is interesting in the fact that author investigated important biological ctivity in detail such as that?it target cell wall-associated molecules, but dysregulation of these molecules leads to enhanced immunomodulatory activity.
However, as a study of antibiotics, this paper lacks detailed information on bioactive substances and is a biological experiment. Therefore, the recommended that this manuscript may be rejected in this time, and this manuscript should be improved according the reviewer’s suggestion before re-submitted.
Major point
1) Although the reviewer understands HPLC fingerprint as fig 1, which showed (b) EEP of Chiang Mai. (c) EEP of Chantaburi,?in this manuscript, the active compounds should be clarified in this?research. The author should isolate the active compound.
2) In figure 2,3 and 4, the positive compounds was not shown. Please discuss the positive?compound suitable in this system.
Author Response
Response to Reviewer 2 Comments
We sincerely appreciate the valuable comments and suggestions from the reviewers. We also thanks for giving us the opportunity to submit a revised of the manuscript. We will detail in our response below for a point-by-point response to the reviewers comments and concerns.
Point 1: Although the reviewer understands HPLC fingerprint as fig 1, which showed (b) EEP of Chiang Mai. (c) EEP of Chantaburi, in this manuscript, the active compounds should be clarified in this research. The author should isolate the active compound.
Response 1: In this research, we used the stingless bee (SLB) propolis from Tetragonula laeviceps and Tetrigona melanoleuca. Both of SLB collected a natural substance from various plants. Since, several studies reported the variable of phytochemical composition in SLB propolis. Previous study characterized the phytochemical compounds in Thai stingless bee propolis by GC/MS analysis [1]. The main constituents of the ethanol extract of Tetrigona melanoleuca propolis were triterpenes. Six prenylated xanthones, one triterpene and one lignane were observed from Tetragonula laeviceps propolis. We expected that phenolic compound, gallic acid, is major composition and flavonoids group as minor composition in both propolis sources to restrict the cryptococcal virulence factors. However, the selected standards for analysis of the active compounds in SLB propolis samples by HPLC have been limited due to propolis is a bee product of plant origin, its chemical composition and biological activity depends on the specificity of the local flora, season of harvest, and bee species [2-4], including the extraction method [5]. Other analytical methods might be included for phytochemical composition analysis in the future.
Point 2: In figure 2, 3 and 4, the positive compounds was not shown. Please discuss the positive compound suitable in this system.
Response 2: Current therapeutic options for cryptococcosis treatment are limited to amphotericin B (AmpB), azoles and 5-flucytosine. Other antifungal drugs, such as nystatin or allylamines, are either not sufficiently absorbed or are too toxic [6]. The mode of action of AmpB binds to ergosterol in the cell membrane, forming pores leading to loss of monovalent cations and subsequent fungal cell death. Azoles inhibit lanosterol 14 α-demethylase, a critical step in ergosterol biosynthesis. In addition, the echinocandins target the fungal cell wall by inhibiting beta-(1,3)-D-glucan synthase [7]. Unlikely, this study focused on of Cryptococcus neoformans virulence factors, especially in chitin/chitosan and melanin, as targets for anticryptococcal treatment [8]. Thus, none of standard antifungal drugs did not include as a positive control due to the mode of action.
References
- Sanpa, S.; Popova, M.; Bankova, V.; Tunkasiri, T.; Eitssayeam, S.; Chantawannakul, P. Antibacterial compounds from propolis of Tetragonula laeviceps and Tetrigona melanoleuca (Hymenoptera: Apidae) from Thailand. PLoS One, 2015, 10, e0126886.
- Popova, M.; Trusheva, B.; Bankova, V. Propolis of stingless bees: A phytochemist's guide through the jungle of tropical biodiversity. Phytomedicine, 2019,153098.
- Saleh, K.; Zhang, T.; Fearnley, J.; Watson, D. A Comparison of the constituents of propolis from different regions of the United Kingdom by liquid chromatography-high resolution mass spectrometry using a metabolomics approach. Curr Metabolomic, 2015, 3, 42-53.
- Przybylek, I.; Karpinski T.M. Antibacterial properties of propolis. Molecules, 2019, 24, 11.
- Trusheva, B.; Trunkova, D.; Bankova, V. Different extraction methods of biologically active components from propolis: a preliminary study. Chem Cent J, 2007, 1, 13.
- Coelho, C.; Casadevall, A. Cryptococcal therapies and drug targets: the old, the new and the promising. Cell Microbiol, 2016, 18, 792-799.
- Wiederhold, N.P. The antifungal arsenal: alternative drugs and future targets. Int J Antimicrob Agents, 2018, 51, 333-339.
- Azevedo, R.; Rizzo, J.; Rodrigues M.L.; Virulence factors as targets for anticryptococcal therapy. J Fungi (Basel), 2016, 2.

Reviewer 3 Report
The article by Mamoon et al entitled “Unveiling the properties of Thai stingless bee propolis via diminishing cell wall-associated cryptococcal melanin and enhancing the fungicidal activity of macrophages” aims at combining the natural power of propolis from bees to damage the cell walls of C. neoformans and increase immune reaction.
The article is well written and of potential interest. No comments on the introduction or materials and methods, but I do have some with respect to results and discussion:
- How variable is the content of phenolic and flavonoid contents of EEPs?
- What’s the vehicle control? Could a positive control also be introduced? Could AMB be used? Could a FICI with AMB be done on the CFU count?
- Can oxidative and nitrosative stress being measured?
- Fig 3 and Fig 4 are way too big, and it is very very difficult to relate their meaning to the text. They should be resized, split in more separate figures and perhaps some of them moved to the supplementary material (for example fig. 4 b, c, d and e, as their results do not display any significant difference with the control). Percentage of phagocytosis and index of phagocytosis…how are they related? Do they come from the same experiment? Are they not a repetition of the same data? And please spend few extra words in explaing the data.
- For figure 4g, what is the percentage of cells in these conditions?
- Sometimes the authors talk of a difference of 8-10%, is this really significant? Normally for a bioactive molecule we would expect more dramatic effects.
Author Response
Response to Reviewer 3 Comments
We sincerely appreciate the valuable comments and suggestions from the reviewers. We also thanks for giving us the opportunity to submit a revised of the manuscript. We will detail in our response below for a point-by-point response to the reviewers comments and concerns. "Please see the attachment."

Round 2
Reviewer 2 Report
The reviewer did not feel improvement of the manuscript.
1) No chemical structure is not suitable to published in the Antibiotics.
2)Please try again the biological experiments including amphotericin B (AmpB), azoles and 5-flucytosine as positive control, even if mechanism is different.
Thus, in this time, this manuscript is not suitable to accept in the Antibiotics.
Reviewer 3 Report
No more comment.